# VARIATIONAL CONTINUAL LEARNING

**Cuong V. Nguyen, Yingzhen Li, Thang D. Bui, Richard E. Turner**
Department of Engineering, University of Cambridge
{vcn22,yl494,tdb40,ret26}@cam.ac.uk

## ABSTRACT

This paper develops variational continual learning (VCL), a simple but general framework for continual learning that fuses online variational inference (VI) and recent advances in Monte Carlo VI for neural networks. The framework can successfully train both deep discriminative models and deep generative models in complex continual learning settings where existing tasks evolve over time and entirely new tasks emerge. Experimental results show that VCL outperforms state-of-the-art continual learning methods on a variety of tasks, avoiding catastrophic forgetting in a fully automatic way.

## 1 INTRODUCTION

Continual learning (also called life-long learning and incremental learning) is a very general form of online learning in which data continuously arrive in a possibly non i.i.d. way, tasks may change over time (e.g. new classes may be discovered), and entirely new tasks can emerge (Schlimmer & Fisher, 1986; Sutton & Whitehead, 1993; Ring, 1997). What is more, continual learning systems must adapt to perform well on the entire set of tasks in an incremental way that avoids revisiting all previous data at each stage. This is a key problem in machine learning since real world tasks continually evolve over time (e.g. they suffer from covariate and dataset shift) and the size of datasets often prohibits frequent batch updating. Moreover, practitioners are often interested in solving a set of related tasks that benefit from being handled jointly in order to leverage multi-task transfer. Continual learning is also of interest to cognitive science, being an intrinsic human ability.

The ubiquity of deep learning means that it is important to develop deep continual learning methods. However, it is challenging to strike a balance between adapting to recent data and retaining knowledge from old data. Too much plasticity leads to the infamous catastrophic forgetting problem (McCloskey & Cohen, 1989; Ratcliff, 1990; Goodfellow et al., 2014a) and too much stability leads to an inability to adapt. Recently there has been a resurgence of interest in this area. One approach trains individual models on each task and then carries out a second stage of training to combine them (Lee et al., 2017). A more elegant and more flexible approach maintains a single model and uses a single type of regularized training that prevents drastic changes in the parameters which have a large influence on prediction, but allows other parameters to change more freely (Li & Hoiem, 2016; Kirkpatrick et al., 2017; Zenke et al., 2017). The approach developed here follows this venerable work, but is arguably more principled, extensible and automatic.

This paper is built on the observation that there already exists an extremely general framework for continual learning: Bayesian inference. Critically, Bayesian inference retains a distribution over model parameters that indicates the plausibility of any setting given the observed data. When new data arrive, we combine what previous data have told us about the model parameters (the previous posterior) with what the current data are telling us (the likelihood). Multiplying and renormalizing yields the new posterior, from which point we can recurse. Critically, the previous posterior constrains parameters that strongly influence prediction, preventing them from changing drastically, but it allows other parameters to change. The wrinkle is that exact Bayesian inference is typically intractable and so approximations are required. Fortunately, there is an extensive literature on approximate inference for neural networks. We merge online variational inference (VI) (Ghahramani & Attias, 2000; Sato, 2001; Broderick et al., 2013) with Monte Carlo VI for neural networks (Blundell et al., 2015) to yield *variational continual learning* (VCL). In addition, we extend VCL to include a small episodic memory by combining VI with the coreset data summarization method (Bachem

et al., 2015; Huggins et al., 2016). We demonstrate that the framework is general, applicable to both deep discriminative models and deep generative models, and that it yields excellent performance.

## 2 CONTINUAL LEARNING BY APPROXIMATE BAYESIAN INFERENCE

Consider a discriminative model that returns a probability distribution over an output $y$ given an input $\boldsymbol{x}$ and parameters $\boldsymbol{\theta}$, that is $p(y|\boldsymbol{\theta}, \boldsymbol{x})$. Below we consider the specific case of a softmax distribution returned by a neural network with weight and bias parameters, but we keep the development general for now. In the continual learning setting, the goal is to learn the parameters of the model from a set of sequentially arriving datasets $\{\boldsymbol{x}_t^{(n)}, y_t^{(n)}\}_{n=1}^{N_t}$ where, in principle, each might contain a single datum, $N_t = 1$. Following a Bayesian approach, a prior distribution $p(\boldsymbol{\theta})$ is placed over $\boldsymbol{\theta}$. The posterior distribution after seeing $T$ datasets is recovered by applying Bayes' rule:

$$p(\boldsymbol{\theta}|\mathcal{D}_{1:T}) \propto p(\boldsymbol{\theta}) \prod_{t=1}^{T} \prod_{n_t=1}^{N_t} p(y_t^{(n_t)}|\boldsymbol{\theta}, \boldsymbol{x}_t^{(n_t)}) = p(\boldsymbol{\theta}) \prod_{t=1}^{T} p(\mathcal{D}_t|\boldsymbol{\theta}) \propto p(\boldsymbol{\theta}|\mathcal{D}_{1:T-1}) p(\mathcal{D}_T|\boldsymbol{\theta}).$$

Here the input dependence has been suppressed on the right hand side to lighten notation. We have used the shorthand $\mathcal{D}_t = \{y_t^{(n)}\}_{n=1}^{N_t}$. Importantly, a recursion has been identified whereby the posterior after seeing the $T$-th dataset is produced by taking the posterior after seeing the $(T-1)$-th dataset, multiplying by the likelihood and renormalizing. In other words, online updating emerges naturally from Bayes' rule.

In most cases the posterior distribution is intractable and approximation is required, even when forming the first posterior $p(\boldsymbol{\theta}|\mathcal{D}_1) \approx q_1(\boldsymbol{\theta}) = \mathrm{proj}(p(\boldsymbol{\theta})p(\mathcal{D}_1|\boldsymbol{\theta}))$. Here $q(\boldsymbol{\theta}) = \mathrm{proj}(p^*(\boldsymbol{\theta}))$ denotes a projection operation that takes the intractable un-normalized distribution $p^*(\boldsymbol{\theta})$ and returns a tractable normalized approximation $q(\boldsymbol{\theta})$. The field of approximate inference provides several choices for the projection operation including i) Laplace's approximation, ii) variational KL minimization, iii) moment matching, and iv) importance sampling. Having approximated the first posterior distribution, subsequent approximations can be produced recursively by combining the approximate posterior distribution with the likelihood and projecting, that is $p(\boldsymbol{\theta}|\mathcal{D}_{1:T}) \approx q_T(\boldsymbol{\theta}) = \mathrm{proj}(q_{T-1}(\boldsymbol{\theta})p(\mathcal{D}_T|\boldsymbol{\theta}))$. In this way online updating is supported. This general approach leads, for the four projection operators previously identified, to i) Laplace propagation (Smola et al., 2004), ii) online VI (Ghahramani & Attias, 2000; Sato, 2001) also known as streaming variational Bayes (Broderick et al., 2013), iii) assumed density filtering (Maybeck, 1982) and iv) sequential Monte Carlo (Liu & Chen, 1998). In this paper the online VI approach is used as it typically outperforms the other methods for complex models in the static setting (Bui et al., 2016) and yet it has not been applied to continual learning of neural networks.

### 2.1 VARIATIONAL CONTINUAL LEARNING (VCL) AND EPISODIC MEMORY ENHANCEMENT

Variational continual learning employs a projection operator defined through a KL divergence minimization over the set of allowed approximate posteriors $\mathcal{Q}$,

$$q_t(\boldsymbol{\theta}) = \arg\min_{q \in \mathcal{Q}} \mathrm{KL}\left(q(\boldsymbol{\theta}) \,\Big\|\, \frac{1}{Z_t} q_{t-1}(\boldsymbol{\theta}) \, p(\mathcal{D}_t|\boldsymbol{\theta})\right), \text{ for } t = 1, 2, \ldots, T. \tag{1}$$

The zeroth approximate distribution is defined to be the prior, $q_0(\boldsymbol{\theta}) = p(\boldsymbol{\theta})$. $Z_t$ is the intractable normalizing constant of $p_t^*(\boldsymbol{\theta}) = q_{t-1}(\boldsymbol{\theta}) \, p(\mathcal{D}_t|\boldsymbol{\theta})$ and is not required to compute the optimum.

VCL will perform exact Bayesian inference if the true posterior is a member of the approximating family, $p(\boldsymbol{\theta}|\mathcal{D}_1, \mathcal{D}_2, \ldots, \mathcal{D}_t) \in \mathcal{Q}$ at every step $t$. Typically this will not be the case and we might worry that performing repeated approximations may accumulate errors causing the algorithm to forget old tasks, for example. Furthermore, the minimization at each step may also be approximate (e.g. due to employing an additional Monte Carlo approximation) and so additional information may be lost. In order to mitigate this potential problem, we extend VCL to include a small representative set of data from previously observed tasks that we call the coreset. The coreset is analogous to an episodic memory that retains key information (in our case, important training data points) from previous tasks which the algorithm can revisit in order to refresh its memory of them. The use of an episodic memory for continual learning has also been explored by Lopez-Paz & Ranzato (2017).

---

**Algorithm 1** Coreset VCL

---

**Input:** Prior $p(\boldsymbol{\theta})$.
**Output:** Variational and predictive distributions at each step $\{q_t(\boldsymbol{\theta}), p(y^*|\boldsymbol{x}^*, \mathcal{D}_{1:t})\}_{t=1}^T$.

   Initialize the coreset and variational approximation: $C_0 \leftarrow \emptyset, \tilde{q}_0 \leftarrow p$.
   **for** $t = 1 \ldots T$ **do**
      Observe the next dataset $\mathcal{D}_t$.
      $C_t \leftarrow$ update the coreset using $C_{t-1}$ and $\mathcal{D}_t$.
      Update the variational distribution for non-coreset data points:

$$\tilde{q}_t(\boldsymbol{\theta}) \leftarrow \arg\min_{q \in \mathcal{Q}} \mathrm{KL}\big(q(\boldsymbol{\theta}) \,\|\, \tfrac{1}{Z}\tilde{q}_{t-1}(\boldsymbol{\theta})\, p(\mathcal{D}_t \cup C_{t-1} \setminus C_t | \boldsymbol{\theta})\big). \tag{2}$$

      Compute the final variational distribution (only used for prediction, and not propagation):

$$q_t(\boldsymbol{\theta}) \leftarrow \arg\min_{q \in \mathcal{Q}} \mathrm{KL}\big(q(\boldsymbol{\theta}) \,\|\, \tfrac{1}{Z}\tilde{q}_t(\boldsymbol{\theta}) p(C_t | \boldsymbol{\theta})\big). \tag{3}$$

      Perform prediction at test input $\boldsymbol{x}^*$: $p(y^*|\boldsymbol{x}^*, \mathcal{D}_{1:t}) = \int q_t(\boldsymbol{\theta}) p(y^*|\boldsymbol{\theta}, \boldsymbol{x}^*)\mathrm{d}\boldsymbol{\theta}$.
   **end for**

---

Algorithm 1 describes coreset VCL. For each task, the new coreset $C_t$ is produced by selecting new data points from the current task and a selection from the old coreset $C_{t-1}$. Any heuristic can be used to make these selections, e.g. $K$ data points can be selected at random from $\mathcal{D}_t$ and added to $C_{t-1}$ to form an unbiased new coreset $C_t$. Alternatively, the greedy $K$-center algorithm (Gonzalez, 1985) can be used to return $K$ data points that are guaranteed to be spread throughout the input space. Next, a variational recursion is developed. Bayes' rule can be used to decompose the true posterior taking care to break out contributions from the coreset,

$$p(\boldsymbol{\theta}|\mathcal{D}_{1:t}) \propto \underbrace{p(\boldsymbol{\theta}|\mathcal{D}_{1:t} \setminus C_t)}_{\substack{\text{posterior from data} \\ \text{not in new coreset}}} \quad \underbrace{p(C_t|\boldsymbol{\theta})}_{\substack{\text{likelihood from data} \\ \text{in new coreset}}} \approx \tilde{q}_t(\boldsymbol{\theta}) p(C_t|\boldsymbol{\theta}).$$

Here the variational distribution $\tilde{q}_t(\boldsymbol{\theta})$ approximates the contribution to the posterior from the non-coreset data points. A recursion is identified by noting

$$p(\boldsymbol{\theta}|\mathcal{D}_{1:t} \setminus C_t) = \underbrace{p(\boldsymbol{\theta}|\mathcal{D}_{1:t-1} \setminus C_{t-1})}_{\substack{\text{previous posterior from} \\ \text{data not in old coreset}}} \underbrace{p(C_{t-1} \setminus C_t | \boldsymbol{\theta})}_{\substack{\text{likelihood from} \\ \text{data leaving coreset}}} \underbrace{p(D_t \setminus C_t | \boldsymbol{\theta})}_{\substack{\text{likelihood from new} \\ \text{data not in new coreset}}} \approx \tilde{q}_{t-1}(\boldsymbol{\theta}) p(\mathcal{D}_t \cup C_{t-1} \setminus C_t | \boldsymbol{\theta}).$$

Hence propagation is performed via $\tilde{q}_t(\boldsymbol{\theta}) = \mathrm{proj}(\tilde{q}_{t-1}(\boldsymbol{\theta}) p(\mathcal{D}_t \cup C_{t-1} \setminus C_t | \boldsymbol{\theta}))$ with VCL employing the variational KL projection. A further projection step is needed before performing prediction $q_t(\boldsymbol{\theta}) = \mathrm{proj}(\tilde{q}_t(\boldsymbol{\theta}) p(C_t | \boldsymbol{\theta}))$. In this way the coreset is incorporated into the approximate posterior directly before prediction which helps mitigate any residual forgetting. From a more general perspective, coreset VCL is equivalent to a message-passing implementation of VI in which the coreset data point updates are scheduled after updating the other data.

## 3    Variational Continual Learning in Deep Discriminative Models

The VCL framework is general and can be applied to many discriminative probabilistic models. Here we apply it to continual learning of deep fully-connected neural network classifiers. Before turning to the application of VCL, we first consider the architecture of neural networks suitable for performing continual learning. In simple instances of discriminative continual learning, where data are arriving in an i.i.d. way or where only the input distribution $p(\boldsymbol{x}_{1:T})$ changes over time, a standard *single-head* discriminative neural network suffices. In many cases the tasks, although related, might involve different output variables. Standard practice in multi-task learning (Bakker & Heskes, 2003) uses networks that share parameters close to the inputs but with separate heads for each output, hence *multi-head* networks. Graphical models depicting the network architecture for deep discriminative and deep generative models are shown in fig. 1.

Recent work has explored more advanced structures for continual learning (Rusu et al., 2016) and multi-task learning more generally (Swietojanski & Renals, 2014; Rebuffi et al., 2017). These architectural advances are complementary to the new learning schemes developed here and a synthesis of the two would be potentially more powerful. Moreover, a general solution to continual learning

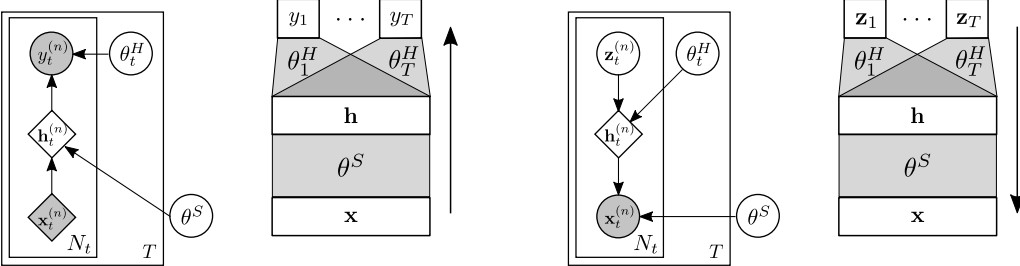

(a) multi-head discriminative network          (b) multi-head generative network

Figure 1: Schematics of the multi-head networks tested in the paper, including both the graphical model (left) and network architecture (right). (a) A multi-head discriminative model showing how network parameters might be shared during training. The lower-level network is parameterized by the variables $\boldsymbol{\theta}^S$ and is shared across multiple tasks. Each task $t$ has its own "head network" $\boldsymbol{\theta}_t^H$ mapping to the outputs from a common hidden layer. The full set of parameters is therefore $\boldsymbol{\theta} = \{\boldsymbol{\theta}_{1:T}^H, \boldsymbol{\theta}^S\}$. (b) A multi-head generative model with shared network parameters. The head networks generate the intermediate level representations from the latent variables $\mathbf{z}$.

would perform *automatic continual model building* adding new bespoke structure to the existing model as new tasks are encountered. Although this is a very interesting research direction, here we make the simplifying assumption that the model structure is known *a priori*.

VCL requires specification of $q(\boldsymbol{\theta})$ where $\boldsymbol{\theta}$ in the current case is a $D$ dimensional vector formed by stacking the network's biases and weights. For simplicity we use a Gaussian mean-field approximate posterior $q_t(\boldsymbol{\theta}) = \prod_{d=1}^D \mathcal{N}(\theta_{t,d}; \mu_{t,d}, \sigma_{t,d}^2)$. Taking the most general case of a multi-head network, before task $k$ is encountered the posterior distribution over the associated head parameters will remain at the prior and so $q(\boldsymbol{\theta}_k^H) = p(\boldsymbol{\theta}_k^H)$. This is convenient as it means the variational approximation can be grown incrementally, starting from the prior, as each task emerges. Moreover, only tasks present in the current dataset $\mathcal{D}_t$ need to have their posterior distributions over head parameters updated. The shared parameters, on the other hand, will be constantly updated. Training the network using the VFE approach in eq. (1) is equivalent to maximizing the negative online variational free energy or the variational lower bound to the online marginal likelihood

$$\mathcal{L}_{\text{VCL}}^t(q_t(\boldsymbol{\theta})) = \sum_{n=1}^{N_t} \mathbb{E}_{\boldsymbol{\theta} \sim q_t(\boldsymbol{\theta})} \left[ \log p(y_t^{(n)}|\boldsymbol{\theta}, \mathbf{x}_t^{(n)}) \right] - \text{KL}(q_t(\boldsymbol{\theta})\|q_{t-1}(\boldsymbol{\theta})) \tag{4}$$

with respect to the variational parameters $\{\mu_{t,d}, \sigma_{t,d}\}_{d=1}^D$. Whilst the KL-divergence $\text{KL}(q_t(\boldsymbol{\theta})\|q_{t-1}(\boldsymbol{\theta}))$ can be computed in closed-form, the expected log-likelihood requires further approximation. Here we take the usual approach of employing simple Monte Carlo and use the *local reparameterization* trick to compute the gradients (Salimans & Knowles, 2013; Kingma & Welling, 2014; Kingma et al., 2015). At the first time step, the prior distribution, and therefore $q_0(\boldsymbol{\theta})$ is chosen to be a multivariate Gaussian distribution (see e.g. Graves, 2011; Blundell et al., 2015).

## 4 VARIATIONAL CONTINUAL LEARNING IN DEEP GENERATIVE MODELS

Deep generative models (DGMs) have garnered much recent attention. By passing a simple noise variable (e.g. Gaussian noise) through a deep neural network, these models have been shown to be able to generate realistic images, sounds and videos sequences (Chung et al., 2015; Kingma et al., 2016; Vondrick et al., 2016). Standard approaches for learning DGMs have focused on batch learning, i.e. the observed instances are assumed to be i.i.d. and are all available at the same time. In this section we extend the VCL framework to encompass variational auto-encoders (VAEs) (Kingma & Welling, 2014; Rezende et al., 2014), a form of DGM. The approach could be extended to generative adversarial networks (GANs) (Goodfellow et al., 2014b) for which continual learning is an open problem (see Seff et al. (2017) for an initial attempt).

Consider a model $p(\mathbf{x}|\mathbf{z}, \boldsymbol{\theta})p(\mathbf{z})$, for observed data $\mathbf{x}$ and latent variables $\mathbf{z}$. The prior over latent variables $p(\mathbf{z})$ is typically Gaussian, and the distributional parameters of $p(\mathbf{x}|\mathbf{z}, \boldsymbol{\theta})$ are defined by a deep neural network. For example, if Bernoulli likelihood is used, then $p(\mathbf{x}|\mathbf{z}, \boldsymbol{\theta}) = \mathrm{Bern}(\mathbf{x}; \boldsymbol{f_\theta}(\mathbf{z}))$, where $\boldsymbol{f_\theta}$ denotes the deep neural network transform and $\boldsymbol{\theta}$ collects all the weight matrices and bias vectors. In the batch setting, given a dataset $\mathcal{D} = \{\mathbf{x}^{(n)}\}_{n=1}^N$, the standard VAE approach learns the parameters $\boldsymbol{\theta}$ by approximate maximum likelihood estimation (MLE). This proceeds by maximizing the variational lower bound with respect to $\boldsymbol{\theta}$ and $\boldsymbol{\phi}$:

$$\mathcal{L}_{\mathrm{VAE}}(\boldsymbol{\theta}, \boldsymbol{\phi}) = \sum_{n=1}^N \mathbb{E}_{q_{\boldsymbol{\phi}}(\mathbf{z}^{(n)}|\mathbf{x}^{(n)})} \left[ \log \frac{p(\mathbf{x}^{(n)}|\mathbf{z}^{(n)}, \boldsymbol{\theta})p(\mathbf{z}^{(n)})}{q_{\boldsymbol{\phi}}(\mathbf{z}^{(n)}|\mathbf{x}^{(n)})} \right], \tag{5}$$

where $\boldsymbol{\phi}$ are the variational parameters of the approximate posterior or "encoder".

The approximate MLE approach is unsuitable for the continual learning setting as it does not return parameter uncertainty estimates that are critical for weighting the information learned from old data. So, instead the VCL approach will approximate the full posterior distribution over parameters, $q_t(\boldsymbol{\theta}) \approx p(\boldsymbol{\theta}|\mathcal{D}_{1:t})$, after observing the $t$-th dataset. Specifically, the approximate posterior $q_t$ is obtained by maximizing the *full* variational lower bound with respect to $q_t$ and $\boldsymbol{\phi}$:

$$\mathcal{L}_{\mathrm{VCL}}^t(q_t(\boldsymbol{\theta}), \boldsymbol{\phi}) = \mathbb{E}_{q_t(\boldsymbol{\theta})} \left\{ \sum_{n=1}^{N_t} \mathbb{E}_{q_{\boldsymbol{\phi}}(\mathbf{z}_t^{(n)}|\mathbf{x}_t^{(n)})} \left[ \log \frac{p(\mathbf{x}_t^{(n)}|\mathbf{z}_t^{(n)}, \boldsymbol{\theta})p(\mathbf{z}_t^{(n)})}{q_{\boldsymbol{\phi}}(\mathbf{z}_t^{(n)}|\mathbf{x}_t^{(n)})} \right] \right\} - \mathrm{KL}(q_t(\boldsymbol{\theta})||q_{t-1}(\boldsymbol{\theta})),$$

where the encoder network $q_{\boldsymbol{\phi}}(\boldsymbol{z}_t^{(n)}|\boldsymbol{x}_t^{(n)})$ is parameterized by $\boldsymbol{\phi}$ which is task-specific. It is likely to be beneficial to share (parts of) these encoder networks, but this is not investigated in this paper.

As was the case for multi-head discriminative models, we can split the generative model into shared and task-specific parts. There are two options: (i) the generative models share across tasks the network that generates observations $\mathbf{x}$ from the intermediate-level representations $\mathbf{h}$, but have private "head networks" for generating $\mathbf{h}$ from the latent variables $\mathbf{z}$ (see fig. 1(b)), and (ii) the other way around. Architecture (i) is arguably more appropriate when data are composed of a common set of structural primitives (such as strokes in handwritten digits) that are selected by high level variables (character identities). Moreover, initial experiments on architecture (ii) indicated that information about the current task tended to be encoded entirely in the task-specific lower-level network negating multi-task transfer. For these reasons, we focus on architecture (i) in the experiments.

## 5 RELATED WORK

**Continual Learning for Deep Discriminative Models:** Many neural network continual learning approaches employ regularized maximum likelihood estimation, optimizing objectives of the form:

$$\mathcal{L}^t(\boldsymbol{\theta}) = \sum_{n=1}^{N_t} \log p(y_t^{(n)}|\boldsymbol{\theta}, \mathbf{x}_t^{(n)}) - \tfrac{1}{2}\lambda_t(\boldsymbol{\theta} - \boldsymbol{\theta}_{t-1})^\mathsf{T}\Sigma_{t-1}^{-1}(\boldsymbol{\theta} - \boldsymbol{\theta}_{t-1}).$$

Here the regularization biases the new parameter estimates towards those estimated at the previous step $\boldsymbol{\theta}_{t-1}$. $\lambda_t$ is a user-selected hyper-parameter that controls the overall contribution from previous data and $\Sigma_{t-1}$ is a matrix (normally diagonal in form) that encodes the relative strength of the regularization on each element of $\boldsymbol{\theta}$. We now discuss specific instances of this scheme:

- **Maximum-likelihood estimation and MAP estimation**: maximum likelihood estimation is recovered when there is no regularization ($\lambda_t = 0$). More generally, the regularization term can be interpreted as a Gaussian prior, $q(\boldsymbol{\theta}|\mathcal{D}_{1:t-1}) = \mathcal{N}(\boldsymbol{\theta}; \boldsymbol{\theta}_{t-1}, \Sigma_{t-1}/\lambda_t)$. The optimization returns the *maximum a posteriori* estimate of the parameters, but this does not directly provide $\Sigma_t$ for the next stage. A simple fix is to set $\Sigma_t = I$ and use cross-validation to find $\lambda_t$, but this approximation is often coarse and leads to catastrophic forgetting (Goodfellow et al., 2014a; Kirkpatrick et al., 2017).

- **Laplace Propagation (LP)** (Smola et al., 2004): applying Laplace's approximation at each step leads to a recursion for $\Sigma_t^{-1}$, which is initialized using the covariance of the Gaussian prior,

$$\Sigma_t^{-1} = \Phi_t + \Sigma_{t-1}^{-1} \text{ where } \Phi_t = -\nabla\nabla_{\boldsymbol{\theta}} \sum_{n=1}^{N_t} \log p(y_t^{(n)}|\boldsymbol{\theta}, \mathbf{x}_t^{(n)})\Big|_{\boldsymbol{\theta}=\boldsymbol{\theta}_t} \text{ and } \lambda_t = 1.$$

  To avoid computing the full Hessian of the likelihood, diagonal Laplace propagation retains only the diagonal terms of $\Sigma_t^{-1}$.

- **Elastic Weight Consolidation (EWC)** (Kirkpatrick et al., 2017) builds on diagonal Laplace propagation by approximating the average Hessian of the likelihoods using well-known identities for the Fisher information:

$$\Phi_t \approx \text{diag} \left( \sum_{n=1}^{N_t} \left( \nabla_{\boldsymbol{\theta}} \log p(y_t^{(n)} | \boldsymbol{\theta}, \mathbf{x}_t^{(n)}) \right)^2 \Big|_{\boldsymbol{\theta}=\boldsymbol{\theta}_t} \right).$$

  EWC also modifies the Laplace regularization, $\frac{1}{2}(\boldsymbol{\theta} - \boldsymbol{\theta}_{t-1})^{\mathsf{T}}(\Sigma_0^{-1} + \sum_{t'=1}^{t-1} \Phi_{t'})(\boldsymbol{\theta} - \boldsymbol{\theta}_{t-1})$, introducing hyper-parameters, removing the prior and regularizing to intermediate parameter estimates, rather than just those derived from the last task, $\frac{1}{2} \sum_{t'=1}^{t-1} \lambda_{t'}(\boldsymbol{\theta} - \boldsymbol{\theta}_{t'-1})^{\mathsf{T}} \Phi_{t'}(\boldsymbol{\theta} - \boldsymbol{\theta}_{t'-1})$. These changes may be unnecessary (Huszár, 2017; 2018) and require storing $\boldsymbol{\theta}_{1:t-1}$, but may slightly improve performance (see Kirkpatrick et al. (2018) and our experiments).

- **Synaptic Intelligence (SI)** (Zenke et al., 2017): SI computes $\Sigma_t^{-1}$ using a measure of the importance of each parameter to each task. Practically, this is achieved by comparing the changing rate of the gradients of the objective and the changing rate of the parameters.

VCL differs from the above methods in several ways. First, unlike MAP, EWC and SI, it does not have free parameters that need to be tuned on a validation set. This can be especially awkward in the online setting. Second, although the KL regularization penalizes the mean of the approximate posterior through a quadratic cost, a full distribution is retained and averaged over at training time and at test time. Third, VI is generally thought to return better uncertainty estimates than approaches like Laplace's method and MAP estimation, and we have argued this is critical for continual learning.

There is a long history of research on approximate Bayesian training of neural networks, including extended Kalman filtering (Singhal & Wu, 1989), Laplace's approximation (MacKay, 1992), variational inference (Hinton & Van Camp, 1993; Barber & Bishop, 1998; Graves, 2011; Blundell et al., 2015; Gal & Ghahramani, 2016), sequential Monte Carlo (de Freitas et al., 2000), expectation propagation (EP) (Hernández-Lobato & Adams, 2015), and approximate power EP (Hernández-Lobato et al., 2016). These approaches have focused on batch learning, but the framework described in section 2 enables them to be applied to continual learning. On the other hand, online variational inference has been previously explored (Ghahramani & Attias, 2000; Broderick et al., 2013; Bui et al., 2017), but not for neural networks or in the context of sets of related complex tasks.

**Continual Learning for Deep Generative Models:** A naïve continual learning approach for deep generative models would directly apply the VAE algorithm to the new dataset $\mathcal{D}_t$ with the model parameters initialized at the previous parameter values $\boldsymbol{\theta}_{t-1}$. The experiments show that this approach leads to catastrophic forgetting, in the sense that the generator can only generate instances that are similar to the data points from the most recently observed task.

Alternatively, EWC regularization can be added to the VAE objective:

$$\mathcal{L}_{\text{EWC}}^t(\boldsymbol{\theta}, \boldsymbol{\phi}) = \sum_{n=1}^{N_t} \mathbb{E}_{q_{\boldsymbol{\phi}}(\mathbf{z}_t^{(n)}|\mathbf{x}_t^{(n)})} \left[ \log \frac{p(\mathbf{x}_t^{(n)}|\mathbf{z}_t^{(n)}, \boldsymbol{\theta})p(\mathbf{z}_t^{(n)})}{q_{\boldsymbol{\phi}}(\mathbf{z}_t^{(n)}|\mathbf{x}_t^{(n)})} \right] - \frac{1}{2} \sum_{t'=1}^{t-1} \lambda_{t'}(\boldsymbol{\theta} - \boldsymbol{\theta}_{t'-1})^{\mathsf{T}} \Phi_{t'}(\boldsymbol{\theta} - \boldsymbol{\theta}_{t'-1}).$$

However computing $\Phi_t$ requires the gradient of the intractable marginal likelihood $\nabla_{\boldsymbol{\theta}} \log p(\mathbf{x}|\boldsymbol{\theta})$. Instead, we can approximate the marginal likelihood by the variational lower bound, i.e.

$$\Phi_t \approx \text{diag} \Big( \sum_{n=1}^{N_t} \Big( \nabla_{\boldsymbol{\theta}} \mathbb{E}_{q_{\boldsymbol{\phi}}(\mathbf{z}_t^{(n)}|\mathbf{x}_t^{(n)})} \Big[ \log \frac{p(\mathbf{x}_t^{(n)}|\mathbf{z}_t^{(n)}, \boldsymbol{\theta})p(\mathbf{z}_t^{(n)})}{q_{\boldsymbol{\phi}}(\mathbf{z}_t^{(n)}|\mathbf{x}_t^{(n)})} \Big] \Big)^2 \Big|_{\boldsymbol{\theta}=\boldsymbol{\theta}_t} \Big).$$

Similar variational lower-bound approximations apply when computing the Hessian matrices for LP and $\Sigma_t^{-1}$ for SI. An importance sampling estimate could also be used (Burda et al., 2016).

## 6  EXPERIMENTS

The experiments evaluate the performance and flexibility of VCL through three discriminative tasks and two generative tasks. Standard continual learning benchmarks are used where possible. Comparisons are made to EWC, diagonal LP and SI that employ tuned hyper-parameters $\lambda$ whereas VCL's objective is hyper-parameter free. More details of the experiment settings and an additional experiment are available in the appendix.[1]

---

[1] An implementation of the methods proposed in this paper can be found at: https://github.com/nvcuong/variational-continual-learning.

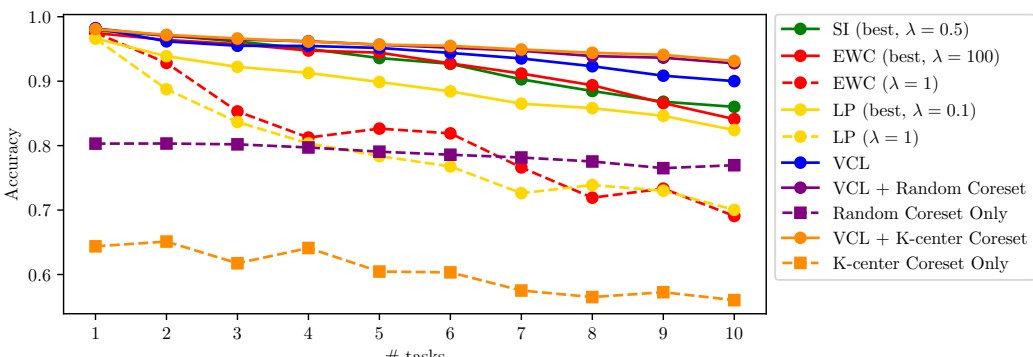

Figure 2: Average test set accuracy on all observed tasks in the Permuted MNIST experiment.

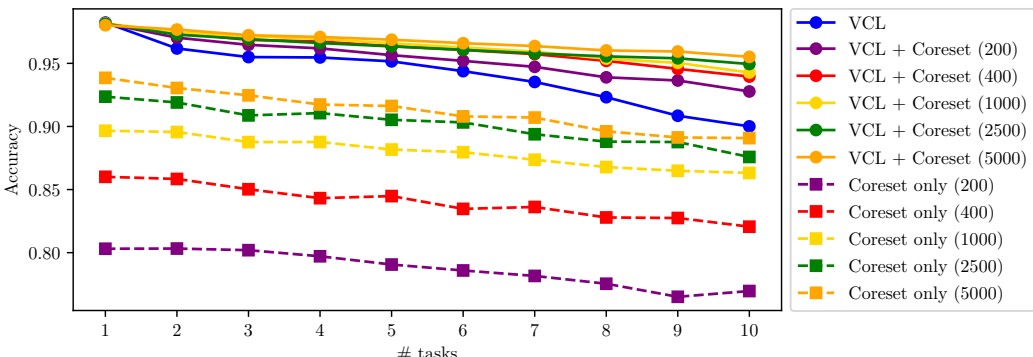

Figure 3: Comparison of the effect of coreset sizes in the Permuted MNIST experiment.

## 6.1 EXPERIMENTS WITH DEEP DISCRIMINATIVE MODELS

We consider the following three continual learning experiments for deep discriminative models.

**Permuted MNIST:** This is a popular continual learning benchmark (Goodfellow et al., 2014a; Kirkpatrick et al., 2017; Zenke et al., 2017). The dataset received at each time step $\mathcal{D}_t$ consists of labeled MNIST images whose pixels have undergone a fixed random permutation. We compare VCL to EWC, SI, and diagonal LP. For all algorithms, we use fully connected single-head networks with two hidden layers, where each layer contains 100 hidden units with ReLU activations. We evaluate three versions of VCL: VCL with no coreset, VCL with a random coreset, and VCL with a coreset selected by the K-center method. For the coresets, we select 200 data points from each task.

Figure 2 compares the average test set accuracy on all observed tasks. From this figure, VCL outperforms EWC, SI, and LP by large margins, even though they benefited from an extensive hyperparameter search for $\lambda$. Diagonal LP performs slightly worse than EWC both when $\lambda = 1$ and when the values of $\lambda$ are tuned. After 10 tasks, VCL achieves 90% average accuracy, while EWC, SI, and LP only achieve 84%, 86%, and 82% respectively. The results also show that the coresets perform poorly by themselves, but combining them with VCL leads to a modest improvement: both random coresets and K-center coresets achieve 93% accuracy.

We also investigate the effect of the coreset size. In fig. 3, we plot the average test set accuracy of VCL with random coresets of different sizes. At the coreset size of 5,000 examples per task, VCL achieves 95.5% accuracy after 10 tasks, which is significantly better than the 90% of vanilla VCL. Performance improves with the coreset size although it asymptotes for large coresets as expected: if a sufficiently large coreset is employed, it will be fully representative of the task and thus training on the coreset alone can achieve a good performance. However, the experiments show that the combination of VCL and coresets is advantageous even for large coresets.

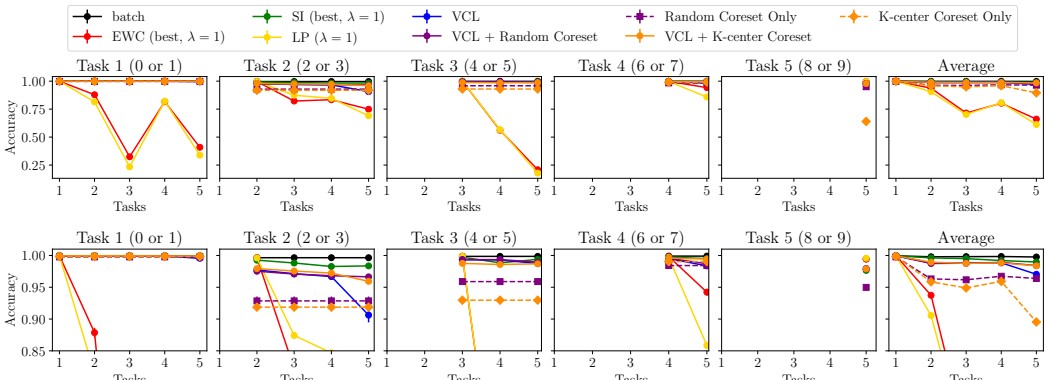

Figure 4: Test set accuracy on all tasks for the Split MNIST experiment. The last column shows the average accuracy over all tasks. The bottom row is a zoomed version of the top row.

**Split MNIST:** This experiment was used by Zenke et al. (2017) to assess the SI method. Five binary classification tasks from the MNIST dataset arrive in sequence: 0/1, 2/3, 4/5, 6/7, and 8/9. We use fully connected multi-head networks with two hidden layers comprising 256 hidden units with ReLU activations. We compare VCL (with and without coresets) to EWC, SI, and diagonal LP. For the coresets, 40 data points from each task are selected through random sampling or the K-center method.

Figure 4 compares the test set accuracy on individual tasks (averaged over 10 runs) as well as the accumulated accuracy averaged over tasks (right). As an upper bound on the algorithms' performance, we compare to batch VI trained on the full dataset. From this figure, VCL significantly outperforms EWC and LP although it is slightly worse than SI. Again, unlike VCL, EWC and SI benefited from a hyper-parameter search for $\lambda$, but a value close to 1 performs well in both cases. After 5 tasks, VCL achieves 97.0% average accuracy on all tasks, while EWC, SI, and LP attain 63.1%, 98.9%, and 61.2% respectively. Adding the coreset improves VCL to around 98.4% accuracy.

**Split notMNIST:** This experiment is similar to the previous one, but it uses the more challenging notMNIST dataset and deeper networks. The notMNIST dataset[2] here contains 400,000 images of the characters from A to J with different font styles. We consider five binary classification tasks: A/F, B/G, C/H, D/I, and E/J using deeper networks comprising four hidden layers of 150 hidden units with ReLU activations. The other settings are kept the same as the previous experiment. VCL is competitive with SI and significantly outperforms EWC and LP (see fig. 5), although the SI and EWC baselines benefited from a hyper-parameter search. VCL achieves 92.0% average accuracy after 5 tasks, while EWC, SI, and LP attain 71%, 94%, and 63% respectively. Adding the random coreset improves the performance of VCL to 96% accuracy.

## 6.2 Experiments with Deep Generative Models

We consider two continual learning experiments for deep generative models: MNIST digit generation and notMNIST (small) character generation. In both cases, ten datasets are received in sequence. For MNIST, the first dataset comprises exclusively of images of the digit zero, the second dataset ones and so on. For notMNIST, the datasets contain the characters A to J in sequence. The generative model consists of shared and task-specific components, each represented by a one hidden layer neural network with 500 hidden units (see fig. 1(b)). The dimensionality of the latent variable $\mathbf{z}$ and the intermediate representation $\mathbf{h}$ are 50 and 500, respectively. We use task-specific encoders that are neural networks with symmetric architectures to the generator.

We compare VCL to naïve online learning using the standard VAE objective, LP, EWC and SI (with hyper-parameters $\lambda = 1, 10, 100$). For full details of the experimental settings see Appendix E. Samples from the generative models attained at different time steps are shown in fig. 6. The naïve

---

[2]The notMNIST dataset is available at: http://yaroslavvb.blogspot.com/2011/09/notmnist-dataset.html.

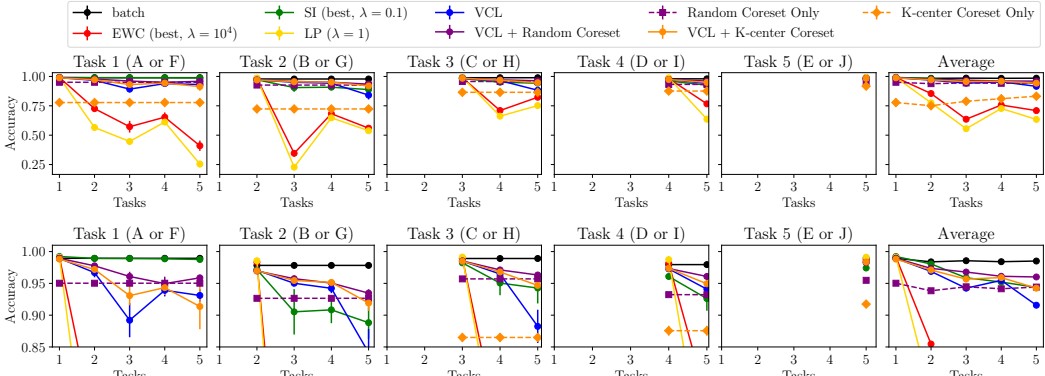

Figure 5: Test set accuracy on all tasks for the Split notMNIST experiment. The last column shows the average accuracy over all tasks. The bottom row is a zoomed version of the top row.

online learning method fails catastrophically and so numerical results are omitted. LP, EWC, SI and VCL remember previous tasks, with SI and VCL achieving the best visual quality on both datasets.

The algorithms are quantitatively evaluated using two metrics in fig. 7: an importance sampling estimate of the test log-likelihood (test-LL) using $5,000$ samples and a measure of quality we term "classifier uncertainty". For the latter, we train a discriminative classifier for the digits/alphabets to achieve high accuracy. The quality of generated samples can then be assessed by the KL-divergence from the one-hot vector indicating the task, to the output classification probability vector computed on the generated images. A well-trained generator will produce images that are correctly classified in high confidence resulting in zero KL. We only report the best performance for LP, EWC and SI.

We observe that LP and EWC perform similarly, most likely due to the fact that both LP and EWC use the same $\Sigma_t$ matrices. EWC achieves significantly worse performance than SI. VCL is on par with or slightly better than SI. VCL has a superior long-term memory of previous tasks which leads to better overall performance on both metrics even though it does not have tuned hyper-parameters in its objective function. For MNIST, the performance of LP and EWC deteriorate markedly when moving from task "digit 0" to "digit 1" possibly due to the large task differences. Also for all experimental settings we tried, SI fails to produce high test-LL results after task "digit 7". Future work will investigate continual learning on a sequence of tasks that follows "adversarial ordering", i.e. the ordering that makes the next task maximally different from the current task.

## 7 CONCLUSION

Approximate Bayesian inference provides a natural framework for continual learning. Variational Continual Learning (VCL), developed in this paper, is an approach in this vein that extends online variational inference to handle more general continual learning tasks and complex neural network models. VCL can be enhanced by including a small episodic memory that leverages coreset algorithms from statistics and connects to message-scheduling in variational message passing. We demonstrated how the VCL framework can be applied to both discriminative and generative models. Experimental results showed state-of-the-art performance when compared to previous continual learning approaches, even though VCL has no free parameters in its objective function. Future work should explore alternative approximate inference methods using the same framework and also develop more sophisticated episodic memories. Finally, we note that VCL is ideally suited for efficient model refinement in sequential decision making problems, such as reinforcement learning and active learning.

### ACKNOWLEDGMENTS

The authors would like to thank Brian Trippe, Siddharth Swaroop, and Matej Balog for insightful comments and discussion. Cuong V. Nguyen is supported by EPSRC grant EP/M0269571. Yingzhen Li is supported by the Schlumberger FFTF Fellowship. Thang D. Bui is supported by the Google European Doctoral Fellowship. Richard E. Turner is supported by Google as well as EPSRC grants EP/M0269571 and EP/L000776/1.

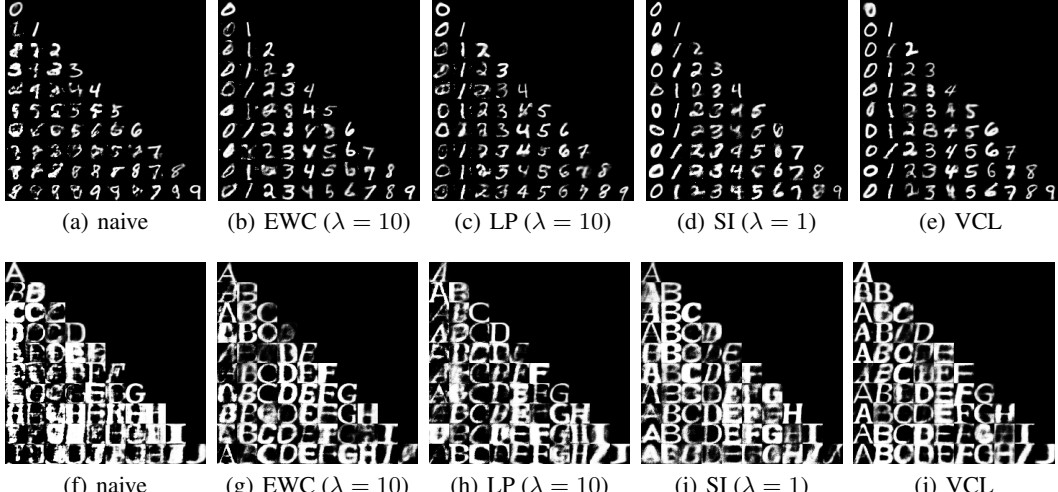

(a) naive    (b) EWC ($\lambda = 10$)    (c) LP ($\lambda = 10$)    (d) SI ($\lambda = 1$)    (e) VCL

(f) naive    (g) EWC ($\lambda = 10$)    (h) LP ($\lambda = 10$)    (i) SI ($\lambda = 1$)    (j) VCL

Figure 6: Generated images from each of the generators after training. Each of the columns shows the images generated from a specific task's generator, and each of the lines shows the generations from generators of all trained tasks. Clearly the naive approach suffers from catastrophic forgetting, while other approaches successfully remember previous tasks.

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

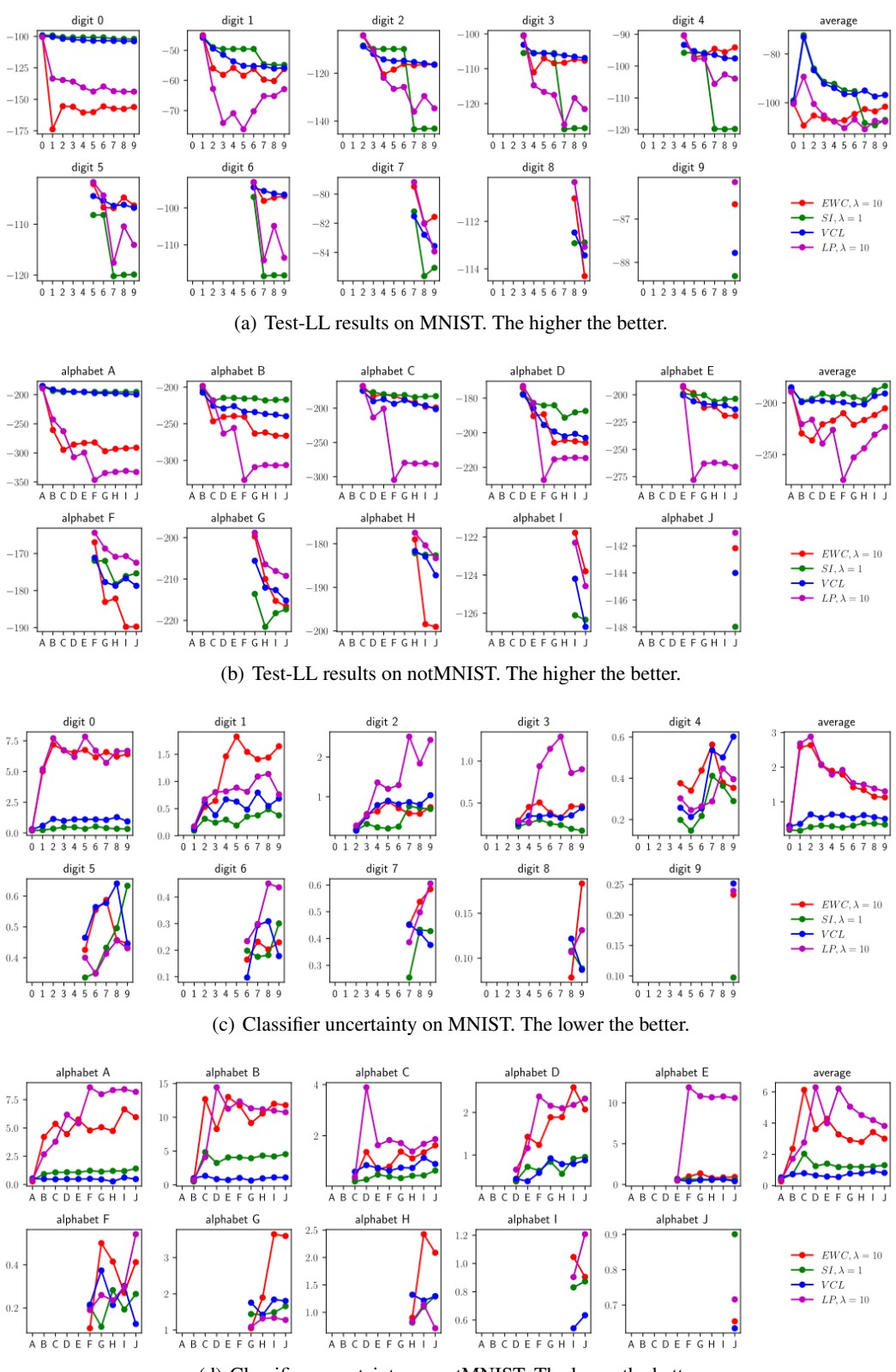

(a) Test-LL results on MNIST. The higher the better.

(b) Test-LL results on notMNIST. The higher the better.

(c) Classifier uncertainty on MNIST. The lower the better.

(d) Classifier uncertainty on notMNIST. The lower the better.

Figure 7: Quantitative results for continual learning for DGMs. See main text for a discussion.

Teofilo F. Gonzalez. Clustering to minimize the maximum intercluster distance. *Theoretical Computer Science*, 1985.

Ian J. Goodfellow, Mehdi Mirza, Da Xiao, Aaron Courville, and Yoshua Bengio. An empirical investigation of catastrophic forgetting in gradient-based neural networks. In *International Conference on Learning Representations*, 2014a.

Ian J. Goodfellow, Jean Pouget-Abadie, Mehdi Mirza, Bing Xu, David Warde-Farley, Sherjil Ozair, Aaron Courville, and Yoshua Bengio. Generative adversarial nets. In *Advances in Neural Information Processing Systems*, 2014b.

Alex Graves. Practical variational inference for neural networks. In *Advances in Neural Information Processing Systems*, 2011.

José Miguel Hernández-Lobato and Ryan P. Adams. Probabilistic backpropagation for scalable learning of Bayesian neural networks. In *International Conference on Machine Learning*, 2015.

José Miguel Hernández-Lobato, Yingzhen Li, Mark Rowland, Daniel Hernández-Lobato, Thang D. Bui, and Richard E. Turner. Black-box $\alpha$-divergence minimization. In *International Conference on Machine Learning*, 2016.

Geoffrey E. Hinton and Drew Van Camp. Keeping the neural networks simple by minimizing the description length of the weights. In *Conference on Computational Learning Theory*, 1993.

Jonathan Huggins, Trevor Campbell, and Tamara Broderick. Coresets for scalable Bayesian logistic regression. In *Advances in Neural Information Processing Systems*, 2016.

Ferenc Huszár. Comment on "Overcoming catastrophic forgetting in NNs": Are multiple penalties needed?, 2017. URL goo.gl/AgdRTN.

Ferenc Huszár. Note on the quadratic penalties in elastic weight consolidation. *Proceedings of the National Academy of Sciences*, 2018.

Diederik Kingma and Jimmy Ba. Adam: A method for stochastic optimization. In *International Conference on Learning Representations*, 2015.

Diederik P. Kingma and Max Welling. Stochastic gradient VB and the variational auto-encoder. In *International Conference on Learning Representations*, 2014.

Diederik P. Kingma, Tim Salimans, and Max Welling. Variational dropout and the local reparameterization trick. In *Advances in Neural Information Processing Systems*, 2015.

Diederik P. Kingma, Tim Salimans, Rafal Jozefowicz, Xi Chen, Ilya Sutskever, and Max Welling. Improved variational inference with inverse autoregressive flow. In *Advances in Neural Information Processing Systems*, 2016.

James Kirkpatrick, Razvan Pascanu, Neil Rabinowitz, Joel Veness, Guillaume Desjardins, Andrei A. Rusu, Kieran Milan, John Quan, Tiago Ramalho, Agnieszka Grabska-Barwinska, Demis Hassabis, Claudia Clopath, Dharshan Kumaran, and Raia Hadsell. Overcoming catastrophic forgetting in neural networks. *Proceedings of the National Academy of Sciences*, 2017.

James Kirkpatrick, Razvan Pascanu, Neil Rabinowitz, Joel Veness, Guillaume Desjardins, Andrei A. Rusu, Kieran Milan, John Quan, Tiago Ramalho, Agnieszka Grabska-Barwinska, Demis Hassabis, Claudia Clopath, Dharshan Kumaran, and Raia Hadsell. Reply to Huszár: The elastic weight consolidation penalty is empirically valid. *Proceedings of the National Academy of Sciences*, 2018.

Sang-Woo Lee, Jin-Hwa Kim, Jung-Woo Ha, and Byoung-Tak Zhang. Overcoming catastrophic forgetting by incremental moment matching. In *Advances in Neural Information Processing Systems*, 2017.

Zhizhong Li and Derek Hoiem. Learning without forgetting. In *European Conference on Computer Vision*, 2016.

Jun S. Liu and Rong Chen. Sequential Monte Carlo methods for dynamic systems. *Journal of the American Statistical Association*, 1998.

David Lopez-Paz and Marc'Aurelio Ranzato. Gradient episodic memory for continual learning. In *Advances in Neural Information Processing Systems*, 2017.

David J.C. MacKay. A practical Bayesian framework for backpropagation networks. *Neural Computation*, 1992.

Peter S. Maybeck. *Stochastic models, estimation, and control*. Academic Press, 1982.

Michael McCloskey and Neal J. Cohen. Catastrophic interference in connectionist networks: The sequential learning problem. *Psychology of Learning and Motivation*, 1989.

Manfred Opper and Cédric Archambeau. The variational Gaussian approximation revisited. *Neural Computation*, 2009.

Roger Ratcliff. Connectionist models of recognition memory: Constraints imposed by learning and forgetting functions. *Psychological Review*, 1990.

Sylvestre-Alvise Rebuffi, Hakan Bilen, and Andrea Vedaldi. Learning multiple visual domains with residual adapters. In *Advances in Neural Information Processing Systems*, 2017.

Danilo J. Rezende, Shakir Mohamed, and Daan Wierstra. Stochastic backpropagation and approximate inference in deep generative models. In *International Conference on Machine Learning*, 2014.

Mark B. Ring. CHILD: A first step towards continual learning. *Machine Learning*, 1997.

Andrei A. Rusu, Neil C. Rabinowitz, Guillaume Desjardins, Hubert Soyer, James Kirkpatrick, Koray Kavukcuoglu, Razvan Pascanu, and Raia Hadsell. Progressive neural networks. *arXiv:1606.04671*, 2016.

Tim Salimans and David A. Knowles. Fixed-form variational posterior approximation through stochastic linear regression. *Bayesian Analysis*, 2013.

Masa-Aki Sato. Online model selection based on the variational Bayes. *Neural Computation*, 2001.

Jeffrey C. Schlimmer and Douglas Fisher. A case study of incremental concept induction. In *The National Conference on Artificial Intelligence*, 1986.

Ari Seff, Alex Beatson, Daniel Suo, and Han Liu. Continual learning in generative adversarial nets. *arXiv:1705.08395*, 2017.

Sharad Singhal and Lance Wu. Training multilayer perceptrons with the extended Kalman algorithm. In *Advances in Neural Information Processing Systems*, 1989.

Alex J. Smola, S.V.N. Vishwanathan, and Eleazar Eskin. Laplace propagation. In *Advances in Neural Information Processing Systems*, 2004.

Richard S. Sutton and Steven D. Whitehead. Online learning with random representations. In *International Conference on Machine Learning*, 1993.

Pawel Swietojanski and Steve Renals. Learning hidden unit contributions for unsupervised speaker adaptation of neural network acoustic models. In *Spoken Language Technology Workshop*, 2014.

Carl Vondrick, Hamed Pirsiavash, and Antonio Torralba. Generating videos with scene dynamics. In *Advances In Neural Information Processing Systems*, 2016.

Friedemann Zenke, Ben Poole, and Surya Ganguli. Continual learning through synaptic intelligence. In *International Conference on Machine Learning*, 2017.

## APPENDIX

### A  FURTHER DETAILS FOR PERMUTED MNIST EXPERIMENT

In this experiment, we use fully connected single-head networks with two hidden layers, where each layer contains 100 hidden units with ReLU activations. The metric used for comparison is the test set accuracy on all observed tasks. We train all the models using the Adam optimizer (Kingma & Ba, 2015) with learning rate $10^{-3}$ since we found that it works best for all models. All the VCL algorithms are trained with batch size 256 and 100 epochs. For all the algorithms with coresets, we choose 200 examples from each task to include into the coresets. The algorithms that use only the coresets are trained using the VFE method with batch size equal to the coreset size and 100 epochs. We use the prior $\mathcal{N}(\mathbf{0}, \mathbf{I})$ and initialize our optimizer for the first task at the mean of the maximum likelihood model and a very small initial variance ($10^{-6}$).

We compare the performance of SI with hyper-parameters $\lambda = 0.01, 0.1, 0.5, 1, 2$ and select the best one ($\lambda = 0.5$) as our baseline (see fig. 8). Following Zenke et al. (2017), we train these models with batch size 256 and 20 epochs. We also compare the performance of EWC with $\lambda = 1, 10, 10^2, 10^3, 10^4$ and select the best value $\lambda = 10^2$ as our baseline (see fig. 9). The models are trained without dropout and with batch size 200 and 20 epochs. We approximate the Fisher information matrices in EWC using 600 random samples drawn from the current dataset. For diagonal LP, we compare the performance of $\lambda = 0.01, 0.1, 1, 10, 100$ and use the best value $\lambda = 0.1$ as our baseline (see fig. 10). The models are also trained with prior $\mathcal{N}(\mathbf{0}, \mathbf{I})$, batch size 200, and 20 epochs. The Hessians of LP are approximated using the Fisher information matrices with 200 samples.

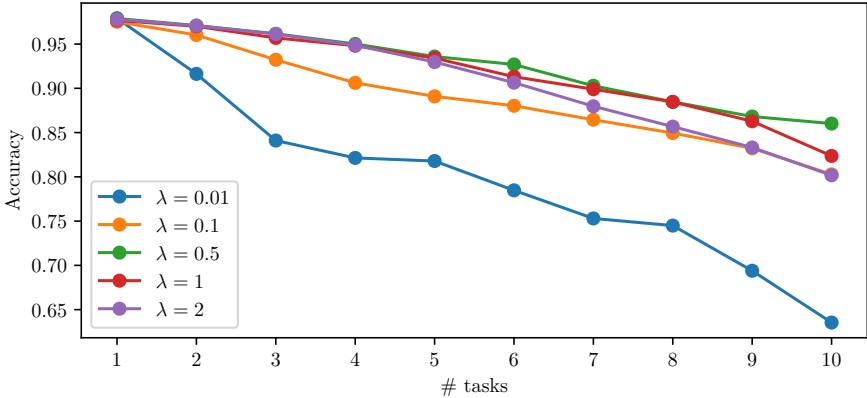

Figure 8: Performance of SI with different hyper-parameter values in Permuted MNIST experiment.

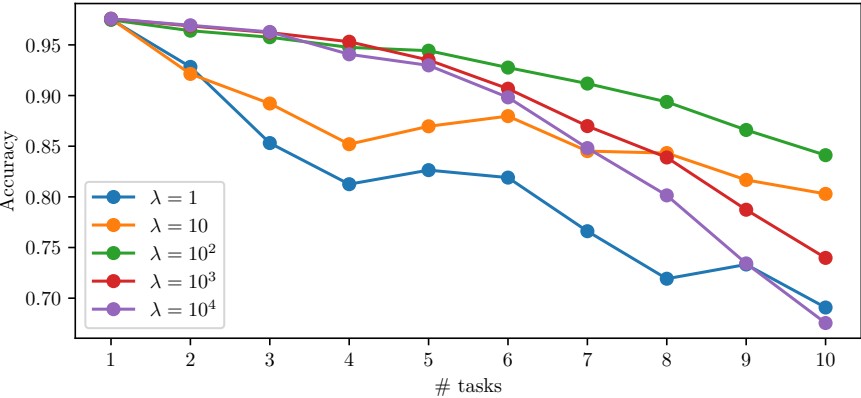

Figure 9: Performance of EWC with different hyper-parameter values in Permuted MNIST experiment.

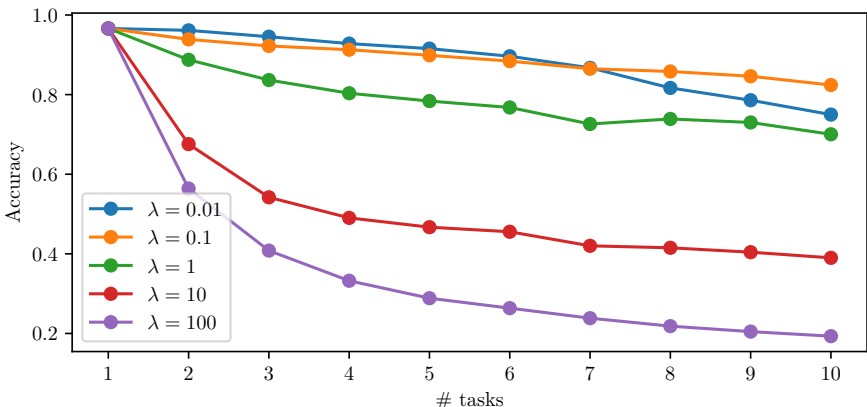

Figure 10: Performance of LP with different hyper-parameter values in Permuted MNIST experiment.

## B    FURTHER DETAILS FOR SPLIT MNIST EXPERIMENT

In this experiment, we use fully connected multi-head networks with two hidden layers, each of which contains 256 hidden units with ReLU activations. At each time step, we compare the test set accuracy of the current model on all observed tasks separately. We also plot the average accuracy over all tasks in the last column of fig. 4. All the results for this experiment are the averages over 10 runs of the algorithms with different random seeds.

We use the Adam optimizer with learning rate $10^{-3}$ for all models. All the VCL algorithms are trained with batch size equal to the size of the training set and 120 epochs. We use the prior $\mathcal{N}(\mathbf{0}, \mathbf{I})$ and initialize our optimizer for the first task at the mean of the maximum likelihood model and a very small initial variance ($10^{-6}$). For the coresets, we choose 40 examples from each task to include into the coresets. In this experiment, the final approximate posterior used for prediction in eq. (3) is computed for each task separately using the coreset points corresponding to the task. The algorithms that use only the coresets are trained using the VFE method with batch size equal to the coreset size and 120 epochs.

We compare the performance of SI with $\lambda = 0.01, 0.1, 1, 2, 3$ and use the best value $\lambda = 1$ as our baseline (see fig. 11). We also compare EWC with both single-head and multi-head models and $\lambda = 1, 10, 10^2, 10^3, 10^4$ (see fig. 12). We approximate the Fisher information matrices using 200 random samples drawn from the current dataset. The figure shows that the multi-head models work better than the single-head models for EWC, and the performance is insensitive to the choice of $\lambda$. Thus, we use the multi-head model with $\lambda = 1$ as the EWC baseline for our experiment. For diagonal LP, we also use the multi-head model with $\lambda = 1$, prior $\mathcal{N}(\mathbf{0}, \mathbf{I})$, and approximate the Hessians using the Fisher information matrices with 200 samples.

## C    FURTHER DETAILS FOR SPLIT NOTMNIST EXPERIMENT

The settings for this experiment are the same as those in the Split MNIST experiment above, except that we use deeper networks with 4 hidden layers, each of which contains 150 hidden units. Figures 13 and 14 show the performance of SI and EWC with different hyper-parameter values respectively. In the experiment, we choose $\lambda = 10^4$ for multi-head EWC, $\lambda = 1$ for multi-head LP, and $\lambda = 0.1$ for SI.

## D    ADDITIONAL EXPERIMENT ON A TOY 2D DATASET

Here we consider a small experiment on a toy 2D dataset to understand some of the properties of EWC with $\lambda = 1$ and VCL. The experiment comprises two sequential binary classification tasks. The first task contains two classes generated from two Gaussian distributions. The data points (with green and black classes) for this task are shown in the first column of fig. 15. The second task contains two classes also generated from two Gaussian distributions. The green class for this task

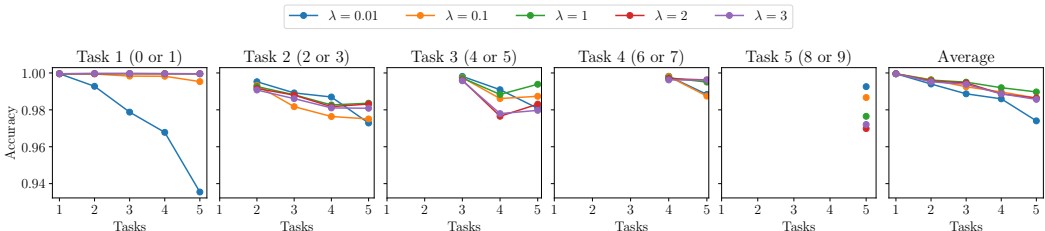

Figure 11: Comparison of SI with different hyper-parameter values in Split MNIST experiment.

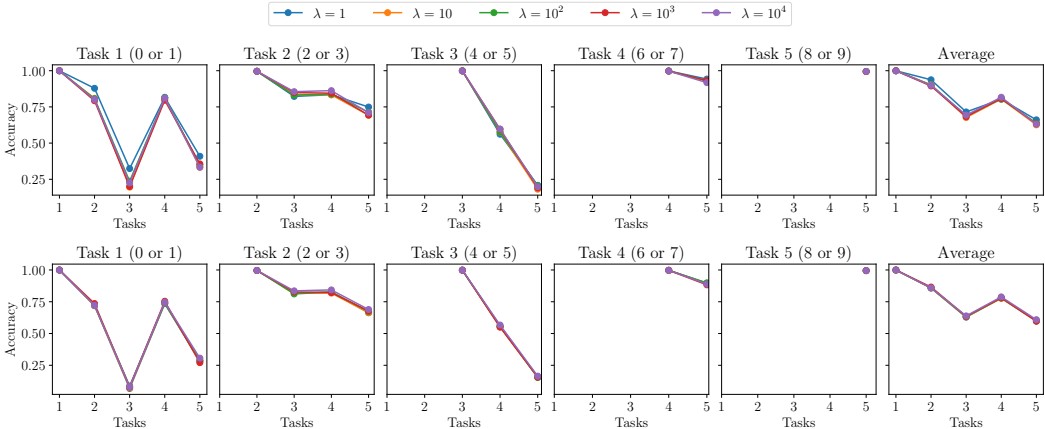

Figure 12: Comparison of EWC with different hyper-parameter values in Split MNIST experiment (top: multi-head model, bottom: single-head model).

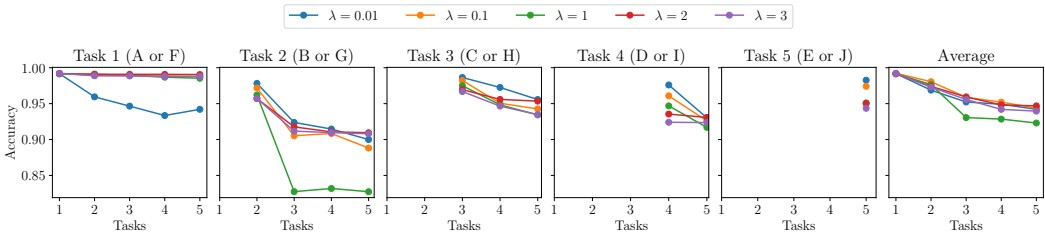

Figure 13: Comparison of SI with different hyper-parameter values in Split notMNIST experiment.

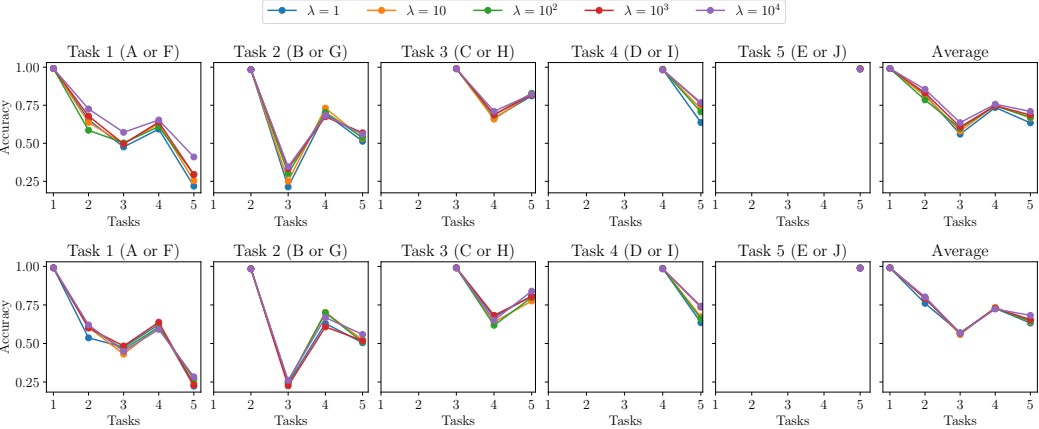

Figure 14: Comparison of EWC with different hyper-parameter values in Split notMNIST experiment (top: multi-head model, bottom: single-head model).

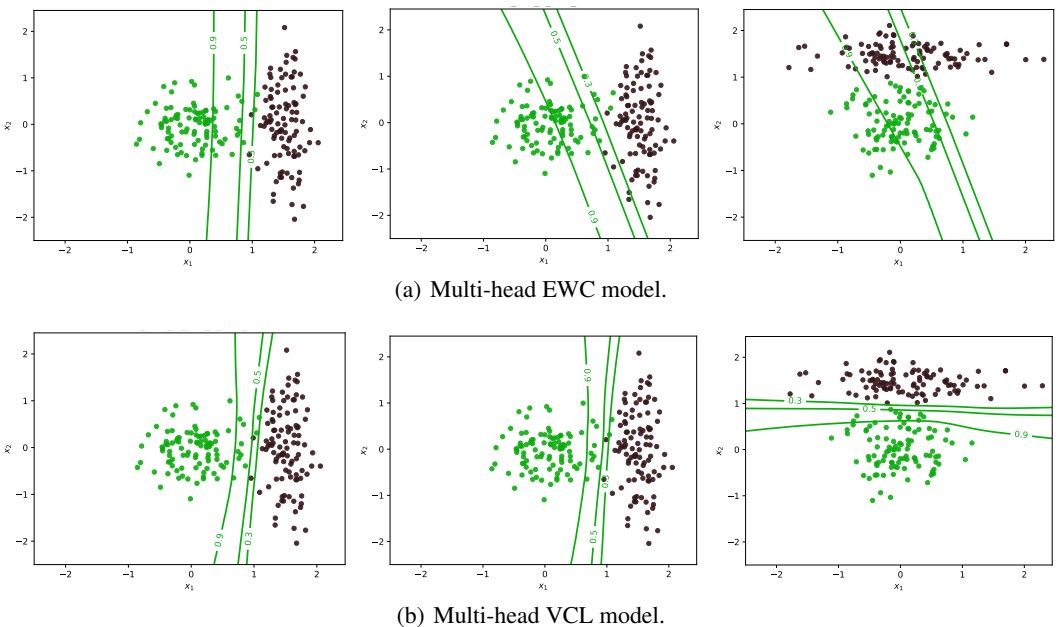

(a) Multi-head EWC model.

(b) Multi-head VCL model.

Figure 15: Comparison of VCL and EWC on a toy 2D dataset. The first column shows the contours of the prediction probabilities after observing the first task. The second and third columns show the contours for the first and the second tasks respectively after observing the second task.

has the same input distribution as the first task, while the input distribution for the black class is different. The data points for this task are shown in the third columns of fig. 15. Each task contains 200 data points with 100 data points in each class.

We compare the multi-head models trained by VCL and EWC on these two tasks. In this experiment, we use fully connected networks with one hidden layer containing 20 hidden units with ReLU activations. The first column of fig. 15 shows the contours of the prediction probabilities after observing the first task, and both methods perform reasonably well for this task. However, after observing the second task, the EWC method fails to learn the classifiers for both tasks, while the VCL method are still able to learn good classifiers for them.

## E    FURTHER DETAILS ON DEEP GENERATIVE MODEL EXPERIMENTS

In the experiments on Deep Generative Models, the learning rates and numbers of optimization epochs are tuned on separate training of each tasks. This gives a learning rate of $10^{-4}$ and the number of epochs 200 for MNIST (except for SI) and 400 for notMNIST. For SI we optimize for 400 epochs on MNIST. For the VCL approach, the parameters of $q_t(\boldsymbol{\theta})$ are initialized to have the same mean as $q_{t-1}(\boldsymbol{\theta})$ and the log standard deviation is set to $10^{-6}$.

The generative model consists of shared and task-specific components, each represented by a one hidden layer neural network with 500 hidden units (see fig. 1(b)). The dimensionality of the latent variable $\mathbf{z}$ and the intermediate representation $\mathbf{h}$ are 50 and 500, respectively. We use task-specific encoders that are neural networks with symmetric architectures to the generator.

## F    MEMORY OF A LINEAR REGRESSION MODEL TRAINED WITH VCL ON RANDOM PATTERNS

In many probabilistic models with conjugate priors, the exact posterior of the parameters/latent variables can be obtained. For example, a Bayesian linear regression model with a Gaussian prior over the parameters and a Gaussian observation model has a Gaussian posterior. If we insist on using a diagonal Gaussian approximation to this posterior and use either the variational free energy method or the Laplace's approximation, we will end up at the same solution – a Gaussian distribution

with the same mean as that of the exact posterior and the diagonal precisions being the diagonal precisions of the exact posterior. Consequently, the online variational Gaussian approximation will give the same result to that given by the online Laplace's approximation. However, when a diagonal Gaussian approximation is used, the batch and sequential solutions are different.

In the following, we will explicitly detail the sequential variational updates for a Bayesian linear regression model to associate *random binary* patterns to *binary* outcomes (Kirkpatrick et al., 2017), and show its relationship to the online Laplace's approximation and the EWC approach of Kirkpatrick et al. (2017). The task consists of associating a random $D$-dimensional binary vector $\mathbf{x}_t$ to a random binary output $y_t$ by learning a weight vector $W$. Note that the possible values of the features and outputs are 1 and $-1$, and not 0 and 1. We also assume that the model sees only one input-output pair $\{\mathbf{x}_t, y_t\}$ at the $t$-th time step and the previous approximate posterior $q_{t-1}(W) = \prod_{d=1}^{D} \mathcal{N}(w_d; m_{t-1,d}, v_{t-1,d})$, and that the observation noise variance $\sigma_y^2$ is fixed. The variational updates for the mean and precisions are available in closed-form as follows:

$$\mathbf{m}_t = \left[\mathbf{I} + \mathbf{V}_{t-1}\frac{\mathbf{x}_t\mathbf{x}_t^{\mathsf{T}}}{\sigma_y^2}\right]^{-1} \left[\mathbf{V}_{t-1}\frac{\mathbf{x}_t y_t^{\mathsf{T}}}{\sigma_y^2} + \mathbf{m}_{t-1}\right], \tag{6}$$

$$v_{t,d}^{-1} = v_{t-1,d}^{-1} + \frac{x_{t,d}^2}{\sigma_y^2} \quad \text{for} \quad d = 1, 2, \ldots, D. \tag{7}$$

By further assuming that $\sigma_y = 1$ and $\|\mathbf{x}_t\|_2 = 1$, the equations above become:

$$\mathbf{m}_t = \left[\mathbf{I} - \frac{\bar{\mathbf{x}}_t\bar{\mathbf{x}}_t^{\mathsf{T}}}{1 + v_0^{-1} + \frac{t}{D}}\right] \mathbf{m}_{t-1} + \frac{\bar{\mathbf{x}}}{1 + v_0^{-1} + \frac{t}{D}}, \tag{8}$$

$$v_{t,d}^{-1} = v_{0,d}^{-1} + \frac{t}{D} \quad \text{for} \quad d = 1, 2, \ldots, D, \tag{9}$$

where $\bar{\mathbf{x}}_t = y_t\mathbf{x}_t$. When $v_{0,d}^{-1} = 0$, i.e. the prior is ignored, the update for the mean above is exactly equation **S4** in the supplementary material of Kirkpatrick et al. (2017). Therefore, in this case, the memory of the network trained by online variational inference is identical to that of the online Laplace's method, provided in (Kirkpatrick et al., 2017). These methods differ, in practice, when the prior is not ignored or when the parameter regularization constraints are accumulated as discussed in the main text. This equivalence also does not hold in the general case, as discussed by Opper & Archambeau (2009) where the Gaussian variational approximation can be interpreted as an *averaged and smoothed* Laplace's approximation.

