# OpenReview forum: "Variational Continual Learning"
_ICLR.cc/2018/Conference — Accept (Poster)_

### Official Review · AnonReviewer3 · 2017-11-24
**This paper introduces a varaitional continual learning framework for neural networks.**

**Rating:** 6
**Confidence:** 3

**Review:**

Overall, the idea of this paper is simple but interesting. Via performing variational inference in a kind of online manner, one can address continual learning for deep discriminative or generative networks with considerations of model uncertainty.

The paper is written well, and literature review is sufficient. My comment is mainly about its importance for large-scale computer vision applications. The neural networks in the experiments are shallow.

---

> ### Author Response · Authors · 2018-01-04
> **Extensions to more complex tasks**
>
> Extensions to more complex tasks:
>
> In the existing discriminative model experiments, we use shallow networks that are comparable to those considered in previous work (Kirkpatrick et al., 2017; Zenke et al., 2017) so that our reimplementation fairly represents the previous work. In the updated version of the paper, we have added an additional Split notMNIST experiment (see page 7 of the new version and Figure 5). The notMNIST dataset is much larger and more noisy than the MNIST dataset. It contains 400,000 images of 10 characters written in different font styles, where each character has 40,000 images. This dataset is considered more difficult than the MNIST dataset. In this new experiment, we investigate a deeper network with 4 hidden layers and our method also performs well compared to EWC and SI.
>
> Extension to computer vision applications:
>
> The paper shows that VCL performs very well for MLPs in a variety of settings which we believe is an important contribution. To apply our method to many large-scale computer vision applications, the method needs to be extended to handle CNNs. In general, accurate approximate variational inference methods have not been developed for CNNs and this is an outstanding goal of the area of Bayesian Deep Learning. We therefore leave this development for future research. However, once a good general variational inference method has been developed for CNNs, it will be straightforward to apply the VCL framework.
>
> Although MC dropout (Gal & Ghahramani, 2016) is one candidate for Bayesian inference in CNNs, the nature of this approximation makes vanilla application of the VCL framework difficult. MC dropout uses a Gaussian prior over the weights and (the limit of) a mixture of Gaussians with shared parameters for the variational distribution. These two distributions are not of the same form and therefore a second approximation step would be required to apply VCL. Moreover, the impoverished representation of posterior uncertainty retained by MC dropout is likely to result in poor continual learning performance since nuanced and parameter specific information about parameter uncertainty is required in this setting. Approximations that employ a single global variance parameter in the q distribution, such as those employed by Kingma et al., 2015, will suffer similar problems.
>
> References:
>
> Y. Gal and Z. Ghahramani. Dropout as a Bayesian Approximation: Representing Model Uncertainty in Deep Learning. ICML 2016.
>
> D.P. Kingma, T. Salimans, M. Welling. Variational Dropout and the Local Reparameterization Trick. NIPS 2015.

---

### Official Review · AnonReviewer2 · 2017-11-27
**Seemingly significant finding, but the title should be rephrased**

**Rating:** 6
**Confidence:** 4

**Review:**

This paper proposes a new method, called VCL, for continual learning. This method is a combination of the online variational inference for streaming environment with Monte Carlo method. The authors further propose to maintain a coreset which consists of representative data points from the past tasks. Such a coreset is used for the main aim of avoiding the catastrophic forgetting problem in continual learning. Extensive experiments shows that VCL performs very well, compared with some state-of-the-art methods.

The authors present two ideas for continual learning in this paper: (1) Combination of online variational inference and sampling method, (2) Use of coreset to deal with the catastrophic forgetting problem. Both ideas have been investigated in Bayesian literature, while (2) has been recently investigated in continual learning. Therefore, the authors seems to be the first to investigate the effectiveness of (1) for continual learning. From extensive experiments, the authors find that the first idea results in VCL which can outperform other state-of-the-art approaches, while the second idea plays little role.

The finding of the effectiveness of idea (1) seems to be significant. The authors did a good job when providing a clear presentation, a detailed analysis about related work, an employment to deep discriminative models and deep generative models, and a thorough investigation of empirical performance.

There are some concerns the authors should consider:
- Since the coreset plays little role in the superior performance of VCL, it might be better if the authors rephrase the title of the paper. When the coreset is empty, VCL turns out to be online variational inference [Broderich et al., 2013; Ghahramani & Attias, 2000]. Their finding of the effectiveness of online variational inference for continual learning should be reflected in the writing of the paper as well.
- It is unclear about the sensitivity of VCL with respect to the size of the coreset. The authors should investigate this aspect.
- What is the trade-off when the size of the coreset increases?

---

> ### Author Response · Authors · 2018-01-04
> **New experiments on coreset sizes and the novelty of VCL**
>
> New experiments showing that coresets can significantly improve VCL's performance:
>
> The use of a coreset can *significantly* improve VCL over the vanilla version. We have added a more comprehensive comparison to the updated version of the paper to make this completely clear (see Figure 3 and the last paragraph on page 6). For example, on the permuted MNIST task when the coreset size is 200 examples per task, the final accuracy of VCL improves from 90% to 93% and when the coreset size is increased to 5,000 examples per task, the performance further improves to 95.5%. These are significant improvements for this dataset. Crucially, using just the coreset alone (and no online inference) still performs significantly worse. Thus, although we agree that VCL alone is effective for continual learning the combination with a coreset can be critical.
>
> Moreover, as now noted in the paper, from a more general perspective, coreset VCL is equivalent to a message-passing implementation of variational inference in which the coreset data point message updates are scheduled last, only after the contributions from other data have been incorporated. This opens the door to versions of VCL which revisit the coreset points several times through learning (rather than just at the end).
>
> Novelty of VCL and contributions of the paper:
>
> The novelty of our VCL method compared to online variational inference (Broderich et al., 2013; Ghahramani & Attias, 2000) is two-fold.
>
> First, online VI has only previously been applied to simple conjugate models. Here instead we consider deep neural networks and variational auto-encoders. Indeed, a Bayesian treatment of the parameters of variational auto-encoders, in addition to the latent variables, is challenging in and of itself. These more complex models require a fusion of online VI and Monte Carlo VI which is technically challenging.
>
> Second, previous work on online VI considers very simple tasks, most typically where the data arrive in iid fashion. Here instead, we consider much more general continual learning tasks that were not previously considered for online VI. The increased inhomogeneity in the data necessitated the development of coreset VI which is more natural and simpler than previous work on coresets for continual learning such as Lopez-Paz and Ranzato (2017) which requires an additional constraint on the optimization objective for every new task.
>
> At a more general level, we also feel that it is important to point out to the continual learning community that standard methods of (approximate) Bayesian inference provide a rich mathematical and algorithmic framework for attacking continual learning that has hitherto been largely overlooked.
>
> Appropriateness of the Title:
>
> Given the two points addressed in the above responses, we believe that the title is appropriate. We have endeavoured to explain the relationship to prior work in the first line of the abstract, “a simple but general framework for continual learning that fuses online variational inference (VI) and recent advances in Monte Carlo VI for neural networks”, which we hope clearly explains the positioning of the paper.

---

### Official Review · AnonReviewer1 · 2017-11-27
**New framework for an important problem, supported with experiments in basic cases.**

**Rating:** 6
**Confidence:** 2

**Review:**

The paper describes the problem of continual learning, the non-iid nature of most real-life data and point out to the catastrophic forgetting phenomena in deep learning. The work defends the point of view that Bayesian inference is the right approach to attack this problem and address difficulties in past implementations.

The paper is well written, the problem is described neatly in conjunction with the past work, and the proposed algorithm is supported by experiments. The work is a useful addition to the community.

My main concern focus on the validity of the proposed model in harder tasks such as the Atari experiments in Kirkpatrick et. al. (2017) or the split CIFAR experiments in Zenke et. al. (2017). Even though the experiments carried out in the paper are important, they fall short of justifying a major step in the direction of the solution for the continual learning problem.

---

> ### Author Response · Authors · 2018-01-04
> **Experiments on Harder Tasks**
>
> New Experiment on a harder task:
>
> In order to further assess the efficacy of VCL on larger scale and more complex tasks we have added an additional experiment to the paper: the new Split notMNIST task on page 7 of the updated paper and Figure 5. The notMNIST dataset is much larger and more noisy than the MNIST dataset. It contains 400,000 images of 10 characters written in different font styles, where each character has 40,000 images. This dataset is generally considered more difficult than the MNIST dataset. In this new experiment, we investigate a deeper network and show that VCL still performs well compared to EWC and SI.
>
> Deployment on tasks requiring CNNs:
>
> The application of VCL to the Atari or Split CIFAR tasks is also a sensible suggestion. However, this requires the development of reliable variational inference methods for convolutional neural networks (CNNs). This is still an outstanding research goal of Bayesian Deep Learning and so we leave this for future research. However, once a good variational inference method has been developed for CNNs, it is straightforward to apply the VCL framework to the above tasks.
>
> Please see more relevant discussions of the points above in the response to Reviewer 3.

---

### Author Response · Authors · 2018-01-04
**Thanks and some clarifications.**

Dear Reviewers,

Many thanks for your detailed reviews. We really appreciate the time and effort you have put into reading and commenting on our paper.

Sorry for not responding to your comments more quickly, but we have been working on a set of new experimental results that have been inspired by your suggestions and which we believe strengthen the paper.

Please also note that there were some errors in the original plots of EWC and K-center Coreset Only methods in Figure 2. We have corrected the plots in our updated paper. The updated results are now consistent with previous findings in Zenke et al. (2017), where EWC and SI are comparable in the Permuted MNIST experiment. The updated results do not change our conclusions in this paper.

We will now address each of your reviews individually.

---

### Decision · Program_Chairs · 2018-01-29
**ICLR 2018 Conference Acceptance Decision**

**Decision:**

Accept (Poster)

**Comment:**

The paper addresses the problem of continual learning and solutions based on variational inference. Updates to the paper have improved it and addresses many of the concerns raised by the reviewers during the discussion period.